# Swine Acute Diarrhea Syndrome Coronavirus: An Overview of Virus Structure and Virus–Host Interactions

**DOI:** 10.3390/ani15020149

**Published:** 2025-01-09

**Authors:** Seung-Hwa Baek, Jung-Eun Park

**Affiliations:** College of Veterinary Medicine, Chungnam National University, Daejeon 34134, Republic of Korea

**Keywords:** swine acute diarrhea syndrome coronavirus, virus-host interactions, cross-species transmission, vaccines, zoonotic threat

## Abstract

Swine acute diarrhea syndrome coronavirus (SADS-CoV) has a high mortality rate in piglets and shows high potential for cross-species transmission, posing a zoonotic threat and a new threat to the swine industry. However, there is currently no vaccine or treatment to prevent this virus, and research on the virus is still in its early stages. In this review, we discuss the structural characteristics of the virus and its interactions with the host. We also identify research trends in therapeutics and vaccines and hope to contribute to a better understanding of SADS-CoVs and to future research.

## 1. Introduction

Coronaviruses (CoVs) are positive-sense, single-stranded RNA viruses belonging to the family *Coronaviridae* and the order *Nidovirales*. These viruses are named for their crown-like structure, which is due to the spike (S) glycoproteins located on the envelope, as confirmed via electron microscope images. The CoV genome is the largest RNA virus, measuring 32 kb in size. Like common RNA viruses, CoVs have a high mutation rate, in combination with their strong tendency to recombine, which results in the infection of new hosts. As a result, CoVs, originating from bats, spread to mammalian and bird hosts, including several livestock; in particular, CoVs spread to humans, as seen with severe acute respiratory coronavirus (SARS-CoV) in 2003, Middle East respiratory syndrome coronavirus (MERS-CoV) in 2012, and SARS-CoV-2 in 2019, all of which caused pandemics [1]. The CoV subfamily is classified into the alpha (α), beta (β), gamma (γ), and delta (δ) genera. Among them, γ- and δ-CoVs infect birds, whereas α- and β-CoVs infect a variety of mammals, including humans.

New swine CoV infections cause enormous economic losses to the global pig industry. For example, a swine CoV infection killed more than 1 million piglets in China in October 2010 [2]. In 2013, swine CoV outbreaks in the United States of America (USA), Canada, and Mexico killed more than 8 million piglets in the USA alone [3]. In the first half of 2017, a swine CoV outbreak in China killed more than 20,000 piglets [4]. The major viruses involved in these infections include infectious gastroenteritis virus (TGEV), porcine delta coronavirus (PDCoV), reemergence porcine epidemic diarrhea virus (PEDV), porcine erythrocyte agglutinating encephalomyelitis virus (PHEV), porcine respiratory coronavirus (PRCV), and swine acute diarrhea syndrome coronavirus (SADS-CoV) [5]. In particular, SADS-CoV occurs more frequently in piglets with acute diarrhea, vomiting, weight loss, and dehydration, resulting in increased mortality and enormous economic losses in the global pig industry [6,7].

The animal coronavirus SADS-CoV poses a persistent threat to human and animal health. However, these threats persist due to the current lack of vaccines or antiviral therapies for SADS-CoV invasion and blockade. Moreover, very little is known about the molecular biology of SADS-CoV. To overcome this challenge, in this review, progress in research on SADS-CoV is summarized, with a focus on the characteristics of the virus, virus-host interactions, therapeutics, and vaccines.

## 2. History of SADS-CoV Outbreaks

In January 2017, a large outbreak of vomiting, dehydration, and diarrheal disease occurred in piglets vaccinated against PEDV on several pig farms in Guangdong Province, southern China. Analysis revealed that bat coronavirus HKU2-derived swine enteric alphacoronavirus (SeACoV) is a new porcine enteric coronavirus (PEC), SADS-CoV. SADS-CoV infection has resulted in the death of approximately 25,000 pigs and considerable economic losses [4,8,9,10]. In 2018, a novel SADS-CoV strain (CH/FJWT/2018; GenBank No. MH615810) was detected in the feces of piglets from seven pig farms in Fujian Province [11]. In February 2019, a diarrheal outbreak that killed 2000 pigs occurred in Guangdong Province, and virological investigation revealed that the causative virus was CN/GDLX/2019 (GenBank No. MK651076) [12]. In 2021, a SADS-CoV outbreak with a 100% mortality rate was identified in Guangxi Province and was designated SADS-CoV/Guangxi/2021 (GenBank No. ON911569) [13]. From 2021 to 2023, Zhang et al. conducted surveillance on pig farms in central China for porcine CoVs and confirmed that SADS-CoV was the cause of death in 400 piglets in Henan Province, Central China. The strain causing the outbreak was designated SADS-CoV/HNNY/2023 (GenBank No. PP069800) [14]. Prior to 2021, SADS-CoV was observed only in coastal China, but the 2023 outbreak in Henan Province strongly suggests that the virus may have spread widely to inland areas of China. As a result of confirming the epidemiological origin of SADS-CoV/HNNY/2023, it belongs to various groups of CoVs found among *Rhinophus*, including HKU2, indicating interspecies transmission events. In addition, phylogenetic analysis of the most variable S gene confirmed that SADS-CoV/HNNY/2023 clusters with viruses from Guangdong Province (Figure 1). These findings suggest that SADS-CoV/HNNY/2023 originated from the ongoing propagation and evolution of SADS-CoV in China rather than from interspecies transmission from bats [14].

Recently, Hassanin et al. reported SADS-CoV-related bat coronaviruses from Vietnam that were sampled from *Rhinolophus affinis* and *Rhinolophus thomasi* [15], suggesting the presence of SADS-CoV outside China. In addition, our group detected SADS-CoV in fecal samples collected from northern Vietnam via multiplex quantitative reverse transcription-polymerase chain reaction (RT-qPCR) [16]. The Vietnamese SADS-CoV S strains had 100% nucleotide and 100% amino acid homology to the SADS-CoV/CN/GDWT/2017 strain (Figure 1). Although how SADS-CoV was introduced in Vietnam is not clear, these results emphasize the importance of biosurveillance to monitor SADS-CoV outside China.

## 3. Virion Structures and Function of SADS-CoV

### 3.1. Genome and Replication Cycle of SADS-CoV

Morphological observations via electron microscopy confirmed that SADS-CoV has a typical crown-like structure, with the S protein distributed on the surface of the viral envelope, similar to other CoVs [17] (Figure 2A). The structure of the SADS-CoV genome contains nine open reading frames (ORFs): ORF1a, ORF1b, S, nonstructural protein 3a (NS3a), envelope (E), membrane (M), nucleocapsid (N), NS7a, and NS7b (Figure 2B). ORF1a and ORF1b, which are located in the 5ʹ region of the genome, encode the functionally conserved replicase–transcriptase complex composed of 16 nonstructural proteins (nsps). The 3′ one-third of the genome encodes four structural proteins (S, E, M, and N), an accessory NS3a between S and E, and two overlapping ORFs (NS7a and NS7b) following the N gene [9].

Genome sequence analysis revealed that SADS-CoV is genetically similar to the bat “HKU2-like-CoV” sublineage. SADS-CoV shares >96% sequence identity with SADS-related coronavirus (SADSr-CoV) detected in various *Rhinolophus* species samples collected in Guangdong Province from 2013 to 2016, further supporting the possibility that SADS-CoV may have originated from bats [18]. Interestingly, SADS-CoV and other HKU2-related α-CoVs share a unique S gene that is closely related to β-CoVs in a manner similar to that of rodent α-CoVs [19]. Although SADS-CoV is classified as an α-CoV, phylogenetic and structural analyses have indicated that the S gene/protein of SADS-CoV is clustered within β-CoVs [8,9]. A structural comparison of the S domains of CoVs from the four genera revealed that the S1 subunit N- and C-terminal domains of HKU2/SADS-CoV are ancestral domains involved in the evolution of CoV S proteins [8]. These data suggest that the S domains of CoVs may have originated through recombination between CoVs and warn of potential recombination events with other human coronaviruses (HCoVs) [9,18].

Viral influx begins with the receptor of the S protein binding to human host cell receptors on the cell surface and the fusion of the viral envelope with the cell membrane. When viral RNA is translated into ORF1a and ORF1b from the cytoplasm of the host cell, two polyproteins, pp1a and pp1ab, are produced, respectively [20]. These polyproteins are cleaved by two virus-encoded protein hydrolases, papain-like protease 2 (PLP2) and 3 C-like proteases (3CLPro), which produce 16 nsps. These nsps form replicase-transcriptase complexes (RTCs), creating suitable environments for RNA synthesis, and are responsible for the replication and transcription of the genomic RNA required for offspring. Moreover, the ORF, located at the 3′ end of the viral genome, encodes important structural proteins, and S, M, and E all migrate to the endoplasmic reticulum (ER)-golf compartment during virus assembly, germination, and transport. The N protein combines with the newly created viral RNA genome to form cytoplasmic nucleoproteins, which are assembled and secreted from the cell through extracellular secretion (Figure 2C) [21].

### 3.2. The Role of SADS-CoV Viral Proteins

The functions of these proteins are summarized in Table 1. Among the 4 structural proteins of CoVs, the S protein is a trimer composed of the S1 and S2 regions and is known to be the main protein that recognizes the receptor for initial entry into the host cell during infection; the S2 region of the S protein fuses with the viral and cell membranes after the S1 region of the S protein binds to the host receptor [17]. 

The S proteins of SARS-CoV, SARS-CoV-2, and HCoV-NL63 bind to the human cell receptor angiotensin-converting enzyme 2 (ACE2) [38,39,40]. In addition, the S protein of MERS-CoV is known to bind to dipeptidyl-peptidase 4 (DPP4) [41], and aminopeptidase N (APN) binds to TGEV, PRCV, and HCoV-229E [42,43]. However, the SADS-CoV S protein does not use these host receptors for viral entry [44]. SADS-CoV has been confirmed to infect various cell lines derived from vertebrates, including humans, confirming the wide host adaptability of SADS-CoV [45,46]. This adaptability has increased the need for research into the human cellular receptors of SADS-CoV, following cases in the past several years of cross-transmission of CoVs from animal reservoirs to humans, resulting in global pandemics [47,48].

The CoV E protein is the smallest and shortest polypeptide among the major structural proteins. The E protein is a multifunctional protein [49], and in addition to its role as a structural protein of the viral capsid, the E protein is involved in viral assembly and budding [50]. Recombinant CoVs lacking E presented reduced viral maturation, markedly reduced titers, and attenuated viral production, demonstrating the importance of E in viral morphogenesis and host affinity [51,52]. The SARS-CoV E protein has been observed in late endosomes and lysosomes within cells and confirmed to be expressed in infected cells. The SADS-CoV E protein also increases membrane permeability to ions and is involved in viral assembly and release [26,27].

The M protein, a membrane-associated, non-glycosylated, and highly conserved protein, is the most abundant essential structural protein in CoVs. The M protein plays an important role in the assembly and release of the virus and the innate immune response of the host through protein-protein interactions with other M protein molecules and other structural proteins, such as the S, E, and N proteins [28]. The M protein has been shown to accumulate in the host nucleus and nucleolus during the early stages of virus infection and to persist in the nucleolus throughout the infection process, where it functions to suppress host cell gene transcription and translation [29]. Like other CoV M proteins, the SADS-CoV M protein plays a significant role in the biological functions of this virus, including viral assembly, budding, and regulation of host innate immunity. In addition, the SADS-CoV M protein has been shown to interact with host proteins in various cellular compartments, including ribosomes, the cytoplasm, and membranes, and to affect host ribose biosynthesis and function, several metabolic pathways, apoptosis, and the phosphatidylinositol 3-kinase (PI3K)/protein kinase B (Akt) signaling pathway [27]. Understanding the virus life cycle by exploring the interactions between the SADS-CoV M protein and host proteins is expected to aid in the development of antiviral drugs and vaccines.

The N protein is the most conserved structural protein and plays a critical role in packaging viral genomic RNA into long-helical ribonucleoprotein (RNP) complexes through interactions with the viral genome and M proteins [30]. The N protein is a highly immunogenic antigen and can be used as a diagnostic antigen and immunogen because it has a low mutation rate and is stable [31,32]. Several studies have also suggested that the N protein is involved in evading the host’s innate immune response [33,34,35]. Since SADS-CoV is a newly emerged CoV, the structural and functional properties of the N protein in SADS-CoV are currently being studied continuously [53].

The various accessory proteins present in CoVs are unique proteins encoded in a genus-specific manner, and the predicted sequences of these proteins do not share a high degree of homology even within the same genus [48]. For example, SARS-CoV has a total of eight accessory proteins, whereas SARS-CoV-2 has nine accessory proteins [54]. The infectious bronchitis virus, which is a γ-CoV, has four accessory proteins, whereas PEDV, which is an α-CoV, has only one accessory protein [55]. The SADS-CoV genome comprises three putative accessory genes (NS3a, NS7a, and NS7b) [27]. Wang et al. reported that SADS-CoV NS7a interacts with apoptosis-inducing factor mitochondria-associated 1 (AIFM1) to activate caspase-3 via caspase-6 and promote SADS-CoV replication in SADS-CoV-infected cells [22]. In addition, SADS-CoV/CN/GDWT/2017 was successfully attenuated by a 58 bp deletion in NS7a/7b through serial passaging in Vero cells, leading to low virulence in piglets [36]. In general, accessory proteins do not affect virus replication *in vitro* but do affect virus pathogenicity and immune modulation *in vivo* [37].

Several viral nsps (nsp1, nsp5, nsp10, nsp12, and nsp16) have been found to inhibit interferon (IFN)-λ1 promoter activity [25]. Recently, SADS-CoV nsp1 blocked Janus kinase 1 (JAK) signal transducer and transcriptional activation factor 1 (STAT1) signaling pathways via the ubiquitin-proteasome pathway [24], thus inhibiting IFN-β production in the host, and SADS-CoV nsp5 targeted mRNA-decapping enzyme 1a (DCP1A) to disrupt type 1 IFN signaling [22]. Transmembrane protein 53 (TMEM53) was found to specifically interfere with viral RNA replication through the inhibition of viral RNA synthesis by inhibiting RNA-dependent RNA polymerase (RdRp) complex assembly through interactions with nsp12 and interfering with the nsp8-nsp12 interaction [56].

## 4. Virus–Host Interactions

### 4.1. Virus-Host Interactions in SADS-CoV Entry

The S protein plays a key role in viral infection by recognizing the cellular receptor and facilitating membrane fusion between the viral envelope and the cell membrane. Several receptors for CoVs, such as ACE2, DPP4, APN, and carcinoembryonic antigen-related cell adhesion molecule 1a (CEACAM1a), have been identified [57]. However, none of these proteins contribute to SADS-CoV infection [45,46], and the host protein partners involved in SADS-CoV infection have yet to be characterized.

The host factors affecting SADS-CoV entry are summarized in Table 2. Chen et al. reported that tunicamycin, an inhibitor of N-linked glycoproteins, blocked the attachment of SADS-CoV to host cells, indicating that SADS-CoV receptors are likely N-linked glycoproteins [58]. Solute carrier family 35 member A1 (SLC35A1), a key component in the sialic acid (SA) synthesis pathway, was identified as a necessary host factor for the infection of swine enteric CoVs, including SADS-CoV [59]. Deletion of SLC35A1 reduced SADS-CoV infectivity, indicating that SLC35A1 may play a role in decreasing viral adsorption to target cells. Wang et al. conducted affinity purification-coupled mass spectrometry to identify host proteins that interact with the SADS-CoV S1 protein [44]. This analysis revealed that peptidylprolyl isomerase B (PPIB) and vimentin are proviral host factors for SADS-CoV infection, although the exact mechanisms by which these factors function remain unclear.

Host proteases that cleave CoV S proteins play critical roles in viral entry [60,61], acting at four distinct stages of the viral infection cycle: (i) proprotein convertases such as furin act during virus packaging; (ii) extracellular proteases such as elastase and exogenous trypsin act after viral release into the extracellular space; (iii) cell surface proteases, including serine proteases such as transmembrane protease serine subtypes (TMPRSSs), act following virus attachment to host cells; and (iv) lysosomal proteases such as cathepsin L and cathepsin B act after virus endocytosis into target cells. Various host proteases have been shown to participate in SADS-CoV entry into different cell types with varying levels of efficiency. For example, furin proteases cleave S proteins at S2/S2 cleavage sites and 97 amino acids upstream, with furin-mediated cleavage linked to the fusogenic properties of S proteins in multiple cell types [47]. Han et al. investigated all 18 members of the type II transmembrane serine protease (TTSP) family via clustered regularly interspaced short palindromic repeats (CRISPR)-based activation of endogenous protein expression and reported that TMPRSS2, TMPRSS4, and TMPRSS13 significantly promote SADS-CoV infection [62]. Additionally, Chen et al. reported that exogenous trypsin, endogenous serine proteases, cathepsin B, and cathepsin L, and lysosomal acidification trigger SADS-CoV entry into cells [58].

In addition, cholic acid (CA) was found to increase SADS-CoV replication in stem cell-derived porcine intestinal enteroids during the early phase of infection [63]. CA triggers several cellular responses, including rapid changes in caveolae-mediated endocytosis, endosomal acidification, and alterations in the endosomal/lysosomal system, all of which are crucial for SADS-CoV entry. Cholesterol 25-hydroxylase (CH25H), a key mediator of innate antiviral immunity, converts cholesterol into 25-hydroxycholesterol (25HC) [64]. 25HC is a soluble factor that regulates sterol biosynthesis by modulating sterol-responsive element-binding proteins (SREBPs) and nuclear receptors [65,66]. Liu et al. demonstrated that CH25H and its enzymatic product, 25HC, inhibit SADS-CoV replication by preventing membrane fusion [67].

**Table 2 animals-15-00149-t002:** Host factors involved in SADS-CoV infection.

Host Factor	Function	Reference
SLC35A1	A key component in the sialic acid Virus adsorption to target cells	[59]
PPIB	Not yet determined	[44]
Vimentin	Not yet determined	[44]
Furin proprotein convertases	S cleavage at and near S1/S2 sites Induce S-mediated cell-cell fusion	[47]
TTSP family (TMPRSS2, TMPRSS4, TMPRSS13)	Trigger SADS-CoV entry into cells	[58,62]
Trypsin	Trigger SADS-CoV entry into cells	[58]
Cathepsin B and cathepsin L	Trigger SADS-CoV entry into cells	[58]
Cholic acid	Enhance SADS-CoV replication by affecting caveolae-mediated endocytosis, endosomal acidification, and alterations in the endosomal/lysosomal system	[63]
CH25H and 25HC	Inhibit SADS-CoV replication by preventing membrane fusion	[67]
PLAC8	Viral trafficking and viral subgenomic RNA expression	[68]
ZDHHC17 (ZD17)	Essential for SADS-CoV genomic RNA replication	[69]
TET2	Induced by type I IFN Inhibit SADS-CoV entry and replication	[70]
RPL18 RALY RHOA	Interact with M protein Affect SADS-CoV replication Not yet determined	[71]
TMEM53	Interact with nsp12 Inhibit RdRp activity and RNA synthesis	[56]
HDAC6	Inhibit SADS-CoV infection by cleaving nsp8 and activating RIG-I-mediated IFN response	[72]

### 4.2. Virus-Host Interactions in SADS–CoV Replication

The host factors affecting SADS-CoV replication are summarized in Table 2.

Tse et al. identified placenta-associated 8 protein (PLAC8) as a critical host factor for SADS-CoV infection via a genome-wide CRISPR knockout screen [68]. Deletion of PLAC8 inhibited viral trafficking and reduced viral subgenomic RNA expression. In a similar approach in which CRISPR knockout screening in HeLa cells was used, Luo et al. identified zinc finger DHHC-type palmitoyltransferase 17 (ZDHHC17 or ZD17) as another key host factor for SADS-CoV infection [73]. Mechanistic studies revealed that ZD17 is essential for SADS-CoV genomic RNA replication. Zeng et al. demonstrated that SADS-CoV induces autophagy by inactivating the protein kinase B (Akt)/ mammalian target of rapamycin (mTOR) pathway and that this autophagy facilitates viral replication [56]. Duan et al. were the first to show that IFN-I inhibits SADS-CoV replication, with tet methylcytosine dioxygenase 2 (TET2) required for this IFN-I-mediated suppression [70]. Through glutathione-S-transferase (GST) pull-down assays combined with liquid chromatography-mass spectrometry (LC-MS/MS), Xu et al. identified 289 host proteins that interact with the SADS-CoV M protein [71]. These proteins are involved in various signaling pathways, including the immune response, apoptosis, ribosome function, and biosynthesis. Among them, ribosomal protein L18 (RPL18), RALY, and ras homolog family member A (RHOA) were found to impact viral replication, although the exact mechanisms involved remain to be elucidated. Yao et al. used large-scale human cDNA screening and discovered that TMEM53 acts as a novel cell-intrinsic restriction factor against SADS-CoV [56]. TMEM53 interacts with nsp12 and disrupts the assembly of the viral RdRp complex by blocking the nsp8-nsp12 interaction, thereby inhibiting RdRp activity and RNA synthesis.

### 4.3. Virus-Host Interactions in Apoptosis and Autophagy

Apoptosis, a form of programmed cell death, is a key mechanism for removing unwanted, damaged, or virus-infected cells [74]. The two main apoptotic pathways are the extrinsic (death receptor) pathway [75] and the intrinsic (mitochondrial) pathway [76]. Key molecules involved in apoptosis include Fas ligand (FasL), caspase-3/6/8/9, apoptosis-inducing factor mitochondrion associated 1 (AIFM1), and Bax, among others [74]. Apoptosis has been observed during the infection cycle of many CoVs [77,78] and is implicated in viral pathogenesis, disease progression, and tissue damage [79,80]. Zhang et al. demonstrated that both the caspase-dependent extrinsic (FasL-mediated) and intrinsic (mitochondria-mediated) apoptotic pathways play central roles in SADS-CoV-induced apoptosis, which facilitates viral replication [81]. Wang et al. screened viral proteins for their ability to induce apoptosis and reported that several nsps, including nsp1, nsp5, nsp6, nsp8, nsp9, nsp14, nsp16, and NS7a, can trigger apoptosis in host cells. Among these proteins, NS7a has the strongest effect on inducing apoptosis, whereas other viral proteins, such as nsp2, nsp3, nsp7, nsp10, nsp12, nsp13, S1, and NS3a, inhibit apoptosis in host cells [22]. NS7a induces apoptosis via the AIFM1-caspase-6 pathway without affecting caspase-8 or caspase-9, suggesting that NS7a mediates apoptosis through AIFM1- and caspase-dependent mechanisms. Additionally, Zhang et al. reported that the extracellular signal-regulated kinase (ERK) signaling pathway is a critical cellular factor that mediates SADS-CoV-induced apoptosis [82]. SADS-CoV activates ERK early in the infection of Vero E6 and IPI-2I cells, and this ERK activation is essential for efficient viral replication *in vitro*, although the specific viral components responsible for this activation remain unclear.

Autophagy is an evolutionarily conserved degradative process that is essential for maintaining cellular homeostasis by removing damaged organelles and long-lived proteins. Some viruses can exploit the autophagy pathway to increase their own replication [83,84]. For SADS-CoV, autophagy facilitates viral replication. The autophagy inducer rapamycin increases SADS-CoV production, whereas the inhibition of autophagy via 3-methyladenine or the blockade of autophagosome-lysosome fusion with bafilomycin A1 suppresses viral replication [69]. Mechanistically, SADS-CoV induces autophagy through the inositol-requiring enzyme 1 (IRE1)–jun N-terminal kinases (JNK)–Beclin1 and AKT/mTOR signaling pathways. Specifically, the PLP2-transmembrane (TM) functional domain of the viral nsp3 protein interacts with glucose/regulated protein 78 (GRP78) to activate the IRE1–JNK–Beclin1 signaling pathway [85]. This interaction inhibits the phosphorylation of the Akt and mTOR proteins [86], reducing their autophagy-suppressive effects [69]. SADS-CoV also promotes autophagy in conjunction with the viral replication transcription complex, leading to increased expression of the autophagy marker microtubule-associated protein 1A/1B-light chain 3 (LC3-II) and an increase in the number of double-membrane vesicle (DMV) structures. In this way, SADS-CoV infection induces autophagy and utilizes these proteins to drive DMV formation, thereby increasing viral replication.

### 4.4. Virus-Host Interactions in Innate Immune Responses

IFN, a key cytokine of the innate immune system, is triggered in response to viral invasion and plays a crucial role in establishing an antiviral state at the site of infection while also regulating the progression of the adaptive immune response. SADS-CoV can counteract IFN production both *in vitro* and *in vivo* [86,87]. The CoV N protein, which is the most abundant viral protein in infected cells shortly after entry, likely plays a significant role in disrupting IFN signaling. Zhou et al. reported that the SADS-CoV N protein inhibits IFN-β production by disrupting the interaction between tumor necrosis factor receptor-associated factor 3 (TRAF3) and TANK binding kinase 1 (TBK1) [35]. Moreover, the N protein interacts with retinoic acid-inducible gene I (RIG-I), independent of its RNA-binding activity, mediating K27-, K48-, and K63-linked ubiquitination of RIG-I, leading to its proteasome-dependent degradation, thus suppressing the host IFN response [34]. The N protein also interacts with the tripartite motif containing 25 (TRIM25) coiled-coil domain (CCD) and RIG-I two tandem caspase activation recruitment domains (2CARDs), inhibiting TRIM25 multimerization and its interaction with RIG-I, thereby suppressing RIG-I signaling and IFN-β production [35]. Additionally, SADS-CoV nsp1 inhibits TBK1 phosphorylation by preventing TBK1 ubiquitin modification, which blocks IFN regulatory factor 3 (IRF3) activation [23]. Nsp1 also disrupts IFN transcriptional enhancer formation by inducing cAMP-response element binding protein (CREB)-binding protein (CBP) degradation and promotes K11/K48-linked polyubiquitination of Janus kinase 1 (JAK1), leading to its degradation via the proteasome pathway. This results in the inhibition of signal transducer and activator of transcription 1 (STAT1) phosphorylation. Furthermore, nsp1 prevents STAT1 acetylation and dephosphorylation by inducing CBP degradation [24]. Another viral protein, SADS-CoV nsp5, interferes with type I IFN signaling by cleaving mRNA-decapping enzyme 1a (DCP1A) [6]. Li et al. demonstrated that histone deacetylase 6 (HDAC6) functions as a broad host restriction factor for PECs, including SADS-CoV, by cleaving nsp8 and activating RIG-I-mediated IFN responses [72]. In response, five different PECs, including SADS-CoV, cleave HDAC6 at Q519, and the resulting cleavage products lose their antiviral activity.

Owing to the high expression of the IFN-λ receptor in epithelial cells, IFN-λ is considered vital for defending against mucosal infections, especially enteric infections [35]. SADS-CoV infection has been shown to suppress the production of IFN-λ [22,25]. Wang et al. reported that the viral protein NS7a inhibited poly(I:C)-induced expression of IFN-λ3 by activating caspase-3, which in turn cleaved IRF3 [22]. Additionally, SADS-CoV nsp1 blocks the activation of the IFN-λ1 promoter, which is mediated by mitochondrial antiviral signaling protein (MAVS), TBK1, IκB kinases epsilon (IKKε), and IRF1 [25].

## 5. Preventive and Control Strategies

### 5.1. Therapeutics

SADS-CoV is a bat-derived virus that has the potential to spread across entire species and has been shown to infect various mammalian cell lines [88]. SADS-CoV can infect cell lines derived from various species, including bats, mice, rats, gerbils, hamsters, pigs, chickens, nonhuman primates, and humans [45,89]. Furthermore, SADS-CoV can replicate effectively in several types of primary human lung cells and primary human intestinal cells [46]. These findings suggest that SADS-CoV has broad cell tropism and the capacity for cross-species transmission. This potential for cross-species transmission has also been confirmed in *in vivo* models. Mei et al. demonstrated that SADS-CoV can infect chickens [90]. While SADS-CoV did not cause visible lesions, it replicated in chicken embryos and induced mild respiratory symptoms in experimentally infected chicks. More importantly, virus shedding and the distribution of SADS-CoV in the lungs, spleen, small intestine, and large intestine of infected chickens were verified through RT-qPCR and immunohistochemical (IHC) staining. Additionally, Chen et al. showed that wild-type BALB/c and C57BL/6J suckling mice less than 7 days old were highly susceptible to SADS-CoV infection via intragastric inoculation, leading to severe illness and death [91]. Similarly, Duan et al. reported that SADS-CoV replicated in neonatal BALB/c mice [92]. SADS-CoV caused severe watery diarrhea, weight loss, and 100% mortality in mice 7 to 14 days after intracerebral infection, and the N protein of the virus was detected in the brain, lungs, spleen, and intestines of the infected mice. Although the exact underlying mechanism remains unclear, these laboratory and *in vivo* studies provide evidence of the interspecies transmissibility and zoonotic potential of SADS-CoV. Notably, infection of cultured human cells has also been confirmed, highlighting the urgent need for vaccine and antiviral drug development. Therefore, research on the development of treatments for SADS-CoV through drug screening has been actively conducted.

Chen et al. screened 3523 compounds to identify those with antiviral activity against SADS-CoV. Gemcitabine, mycophenolate mofetil (MMF), mycophenolic acid (MPA) and methylene blue have all been shown to inhibit viral replication after the entry of SADS-CoV. Cepharanthine and methylene blue have been confirmed as blockers of the entry phase of SADS-CoV. MMF and MPA, immunosuppressants with similar structures, inhibit SADS-CoV replication and viral progeny production [93]. Emodin, the main component of aloe extract, has been shown to have antiviral activity throughout the SADS-CoV replication cycle. Emodin primarily reduces the attachment of viral particles to the cell surface, and its antiviral activity has been confirmed to involve the activation of the toll-like receptor 3-IFN-λ3-IFN-stimulated gene 15 pathway in certain cells, thereby modulating the immune response of the host cell [94]. Another natural product, gossypol, was confirmed to have an antiviral effect against SADS-CoV by inhibiting RdRp, a key enzyme involved in viral replication [95]. Quercetin has been shown to be an effective inhibitor of intracellular SADS-CoV proliferation, targeting the adsorption and replication phases of the virus life cycle, reducing the expression level of SADS-CoV-infected piglet intestinal inflammatory factors and reducing pathological damage in *in vivo* experiments [96]. Zhang et al. confirmed that IFN-δ8 reduces SADS-CoV proliferation in swine testicular (ST) cells. Additional *in vivo* experiments demonstrated that the intraperitoneal injection of IFN-δ8 into piglets attenuated intestinal damage and decreased the viral load in the jejunum and ileum [97]. Su et al. screened small-molecule drugs targeting 3CLpro via molecular docking and reported that octyl gallate (OG), a widely used food additive, exhibited strong binding affinity to the 3CLpro active site. 3CLpro is involved in the cleavage of viral polyproteins, suppresses the host antiviral response, and plays a critical role in viral replication, making it a prime target for the development of broad-spectrum anti-CoV drugs. Furthermore, OG strongly inhibited the replication of TGEV, SADS-CoV, and PDCoV *in vitro* and was verified to protect piglets from PEDV infection *in vivo* [98]. RNA interference (RNAi), a process by which gene expression is controlled via a very precise mechanism of sequence-directed gene silencing via short hairpin RNA (shRNA), has quickly emerged as a novel therapeutic approach. Li et al. researched a small interfering RNA generation system in which two different shRNAs targeting the N gene of PDCoV and the M genes of PEDV and SADS-CoV were expressed. The expression of these specific shRNA molecules strongly inhibited the expression of gene RNA in infected cell cultures while simultaneously impairing the replication of the virus. Therefore, this RNAi-based technology is a novel research approach for the treatment and prevention of various viral infections, including SADS-CoV infection [99]. Zhou and Zhang et al. successfully generated and purified six monoclonal antibodies (mAbs) specifically targeting the S protein of SADS-CoV, three of which confirmed the neutralizing activity of SADS-CoV infection in HeLa-R19 and A549 cells and inhibited human-mouse erythrocyte aggregation via two other antibodies. These antibodies have strong potential as therapeutic agents and vaccines against SADS-CoV infection [100]. Zhang et al. confirmed the production of 5D6, a hybridoma cell line that secretes SADS-CoV-specific neutralizing antibodies, and identified the SADS-CoV-derived epitope, the 136-STHAAD-142 region, which does not cross-react with other PECs. SADS-CoV could be used to identify only SADS-CoV strains, demonstrating that SADS-CoV is suitable for diagnostic and therapeutic tool development [101].

### 5.2. Vaccines

The emergence of novel CoVs in both humans and animals underscores the need to prevent potential outbreaks of animal and human diseases [102]. *In vitro* studies have shown that SADS-CoV can infect cell lines of various species, including humans, and poses a potential risk to human health [89]. Rapid response to these outbreaks and the development of effective countermeasures requires the study of antiviral activity and vaccine development. Vaccination is the best way to prevent viral infection, but there are currently no commercially available vaccines that prevent SADS-CoV infection.

Considering the high pathogenicity of SADS-CoV, there is an urgent need to isolate attenuated strains and develop efficient vaccines to control SADS-CoV. Sun et al. first isolated attenuated SADS-CoV/GDWT-P83 by serially propagating the virulent strain SADS-CoV/CN/GDWT/2017 up to 83 times in Vero cells and identified the genetic and pathogenic properties of this strain. After inoculation with SADS-CoV/GDWT-P83, which was confirmed to be attenuated, mild and transient diarrhea was observed in neonatal piglets, confirming a decrease in pathogenicity. These findings suggest that the SADS-CoV/GDWT-P83 strain could be a potential attenuated vaccine candidate against SADS-CoV infection [36]. Zhu et al. generated a replication-competent vesicular stomatitis virus (VSV)-Venus-SADS S virus by replacing the native glycoprotein of VSV with a SADS-CoV S protein and inserting a Venus reporter in the 3′ leader region. Furthermore, we identified potent neutralizing antibodies against SADS-CoV in mice vaccinated with rVSV-Venus-SADS SΔ11 [103].

## 6. Conclusions and Perspectives

CoVs continue to pose a significant threat to human society, showing the ability to cross species barriers and substantially impact human health. In the past, swine CoVs presented a low risk of direct human infection, primarily causing economic losses in the swine industry. However, the recently identified SADS-CoV has been shown to infect various animal cells, including human cells, raising concerns about possible cross-species transmission. While the genomic, structural, evolutionary, and epidemiological characteristics of SADS-CoV have been examined in previous studies, there is still limited information regarding the molecular biology of this virus, such as its receptors and pathogenic mechanisms. The absence of a known receptor, along with the lack of therapeutic options and vaccines, underscores the need for ongoing research to mitigate the risks associated with this virus. To promote SADS-CoV research and establish effective countermeasures, identifying cellular receptors and cofactors is important. Although pathogenicity has been confirmed in some animals, considering the ability to infect various cells, it is also necessary to evaluate pathogenicity in more animals, including livestock and companion animals, to explore the possibility of interspecies transmission. In addition to the identified host factors, identifying more host factors involved in viral infection and pathogenicity is also important for understanding the virus and establishing treatments. Finally, research on immune responses in infected and vaccinated animals is urgently needed.

In this systematic review, findings from existing studies on the molecular biology of SADS-CoV and its interactions with hosts since the virus was first reported are compiled, and comprehensive insights that can guide future research are provided. In addition, these findings highlight the critical need for further investigations to address the challenges posed by this CoV effectively.

## Figures and Tables

**Figure 1 animals-15-00149-f001:**
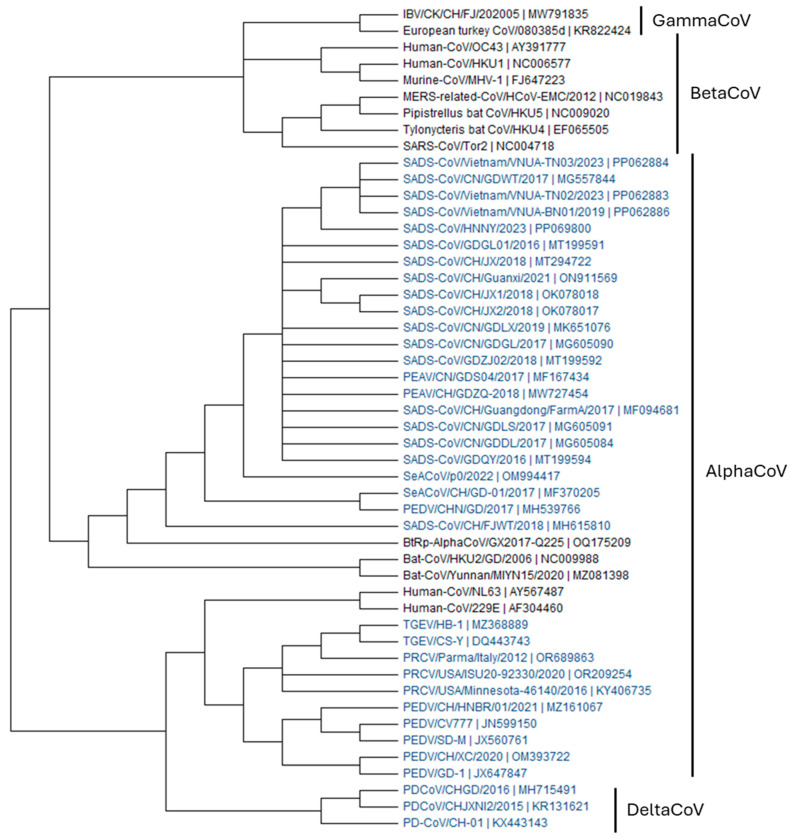
Phylogenetic analysis of published SADS-CoV sequences. The data in the phylogenetic tree come from the full-length S sequence of the virus in national center for biotechnology information (NCBI) and the tree was drawn using Molecular Evolutionary Genetics Analysis (MEGA) software (Version 11.0). Swine CoVs were highlighted in blue.

**Figure 2 animals-15-00149-f002:**
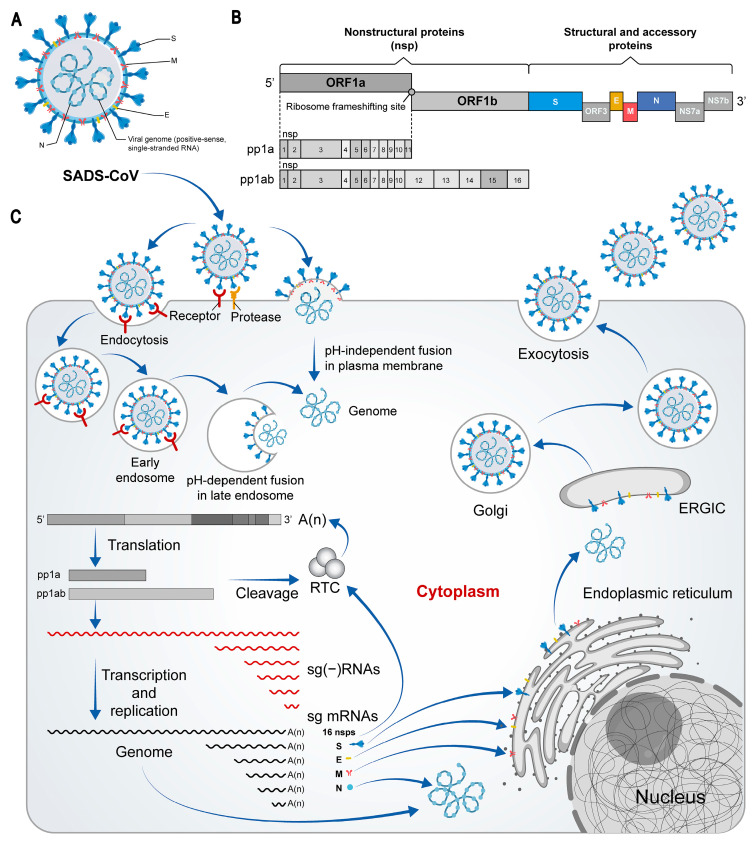
Schematic representation of the SADS structure and life cycle. (**A**) Structure of SADS-CoV. Four structural proteins (S, E, M, N) and RNA are indicated. (**B**) SADS-CoV genomic structure. The viral RNA contains nine open reading frames (ORFs): ORF1a, ORF1b, S, NS3a, E, M, N, NS7a, and NS7b. ORF1a and ORF1b, which are located in the 5ʹ region of the genome, encode the functionally conserved replicase–transcriptase complex composed of 16 nonstructural proteins (nsp1-16). (**C**) Life cycle of SADS-CoV in host cells. The S protein initiates infection by binding to the host cell receptor and facilitating the fusion of the viral envelope with the cell membrane. Once inside, the viral RNA is released into the cytoplasm of the host cell, where ORF1a and ORF1b are translated into polyproteins (pp1a and pp1ab). These polyproteins are subsequently cleaved by the viral proteases 3CLpro and PLpro into 16 nonstructural proteins (nsps). Together, these nsps form a replication and transcription complex responsible for synthesizing the viral genomic RNA needed for progeny. Concurrently, the ORFs at the 3′-end are translated into structural proteins, and the S, M, and E proteins are transported to the ER-Golgi compartment. They assemble with genomic RNA containing the N protein and are eventually secreted from the cell via exocytosis.

**Table 1 animals-15-00149-t001:** The role of SADS-CoV viral proteins.

Viral Proteins	Function	Reference
nsp1	Inhibit TBK1 phosphorylation Disrupt IFN transcriptional enhance formation Prevent STAT1 acetylation and dephosphorylation Block the activation of the IFN-λ1 promoter Trigger apoptosis in host cells	[22], [23], [24], [25]
nsp2	Inhibit apoptosis in host cells	[22]
nsp3	Papain-like protease Inhibit apoptosis in host cells	[22]
nsp5	3-chymotrypsin-like protease Interfere type I IFN signaling by cleaving DCP1A Trigger apoptosis in host cells	[6], [22]
nsp6	Trigger apoptosis in host cells	[22]
nsp7	Inhibit apoptosis in host cells	[22]
nsp8	Trigger apoptosis in host cells	[22]
nsp9	Trigger apoptosis in host cells	[22]
nsp10	Inhibit apoptosis in host cells	[22]
nsp12	RNA-dependent RNA polymerase Inhibit apoptosis in host cells	[22]
nsp13a	Inhibit apoptosis in host cells	[22]
nsp14	Trigger apoptosis in host cells	[22]
nsp16	Trigger apoptosis in host cells	[22]
S	Mediate virus entry by recognizing host receptor(s) and mediating membrane fusion Inhibit apoptosis in host cells (S1)	[22]
NS3a	Inhibit apoptosis in host cells	[22]
E	Involved in viral assembly and release Increase membrane permeability to ions	[26], [27]
M	Involved in viral assembly and release Induce innate immune response Suppress host cell gene transcription and translation Interact with host proteins in various cellular compartments Affect host ribose biosynthesis, metabolic pathways, apoptosis, and PI3K-Akt signaling pathway	[27], [28], [29]
N	Packaging viral RNA into RNP complexes Interact with viral RNA and M protein Highly immunogenic (a potential target for diagnosis and vaccine) Suppress the host innate immune response	[30,31,32,33,34,35]
NS7a/7b	Induce apoptosis via the AIFM1-caspase-6 pathway Inhibit poly(I:C)-induced expression of IFN-λ3 Enhance SADS-CoV replication and virulence Affect virus pathogenicity and immune modulation *in vivo*	[23], [36], [37]

## Data Availability

No new data were created or analyzed in this study. Data sharing is not applicable to this article.

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
