# Peer review of "Swine Acute Diarrhea Syndrome Coronavirus: An Overview of Virus Structure and Virus–Host Interactions"

_animals, 2025, doi:10.3390/ani15020149_

Round 1

Reviewer 1 Report

Comments and Suggestions for Authors

This review discusses the structural characteristics of SADS-CoV and its interactions with the host. Additionally, it identifies research trends in therapeutics and vaccines, aiming to contribute to a better understanding of the virus and to guide future research. Although the manuscript presents a clear, logical structure, it lacks depth in its insights.

1. The recommended revision for Section 2.1 is "The Genome and Replication Cycle of SADS-CoV."

2. The roles of the non-structural proteins of SADS-CoV should be introduced and emphasized, as they play a crucial role in the virus's infection process.

3. Please revise the subtitle of the third section, as it focuses on the interaction between the virus and the host rather than emphasizing the role of host factors in the viral replication cycle.

4. The fourth section recommends including an introduction that outlines the research progress on SADS-CoV vaccines.

5. The author is encouraged to summarize and discuss how SADS-CoV regulates adaptive immune responses, particularly cellular immunity, as this understanding is essential for elucidating the pathogenesis of SADS-CoV and for informing vaccine development.

6. It is highly advisable for the author to create a diagram illustrating the interactions between the SADS-CoV-encoded proteins and host molecules, as this would enhance reader comprehension and serve as a significant highlight of the article.

Reviewer 2 Report

Comments and Suggestions for Authors

Peer review:

Manuscript “Swine acute diarrhea syndrome coronavirus: An updated overview of virus‒host interactions” (animals-3364235)

Review comments to the authors:

This manuscript submitted to Animals is a review on the swine acute diarrhea syndrome coronavirus (SADS-CoV). SADS-CoV induces diarrhea and weight loss in infected piglets. This virus can infect different animal cell lines, showing significant potential for cross-species transmission and representing a possible zoonotic threat. The review focuses on the characteristics of the virus and its interactions with the host. It also addresses potential therapeutic molecules and vaccines to prevent SADS-CoV infection.

In my opinion, the manuscript presents a very good description of the molecular structure of SADS-CoV, as well as its interactions with the host cell. However, I also think that the manuscript is incomplete. I recommend writing a specific section on the epidemiology of SADS-CoV, detailing the dissemination and molecular characterization of the different strains involved in previous outbreaks (i.e. writing in more detail and in a separate section what is summarized in the second paragraph).

There are also needs for minor adjustments. I am detailing some of the main necessary modifications below:

1)    It is necessary to prepare better the Abstract! It is too summarized. It is not presenting all the main topics presented in the review.

2)    The Introduction should describe the clinical impact of the disease in China and worldwide. Please explain the importance of the disease and the main clinical consequences for the animals and the swine industry in this topic. In addition, transfer the epidemiology to a separate section in the main body of the manuscript.

3)    The tables and figures are important and interesting. However, it is not necessary to include tables with the same topics described in the text (as Tables 1 and 2). I think it would be more interesting to present the main viral proteins in a Figure. I also suggest presenting a phylogeny with the characterized strains of the different outbreaks.

4)    As the Abstract describes the manuscript focus on the developments in therapeutics and vaccines, I suggest the authors to describe “therapeutics” and “vaccines” in two separate sections in the body of the manuscript. These sections could be enriched with more recent information.

5)    It is necessary to review the entire text to use the correct scientific nomenclature (remember that bacterial names must always be written in italics).

Finally, the authors have to review all these sections to remove repetition of information.  After all this, the article can be sent for a new peer review.

Reviewer 3 Report

Comments and Suggestions for Authors

The authors present a review on the swine acute diarrhea syndrome coronavirus.  This disease is described to cause moderate to severe diarrhea in piglets with high mortalities.  The authors provide a comprehensive review of the virus structure and virus host interactions at the cellular level.  They also express concern that this bat derived coronavirus has the potential to infect many different species of birds and animals as well as human derived cell cultures.

This is a very comprehensive review comparing the swine acute diarrhea syndrome CoV to the other classes of coronavirruses.  At times in the review if is not clear whether the authors are referring to this coronavirus or other coronaviruses.  Authors it is worth reviewing to be sure you emphasize the swine acute diarrhea virus.  Also, refer to your own studies, this is also not clear or well defined in the review.  Furthermore, can you suggest some next steps where plausible, rather than just stating there is a need for more research?  What questions are unanswered?

Finally, the title does not represent the content of the paper.  Suggest "Swine acute diarrhea syndrome coronavirus:  An overview of virus structure and virus host interactions."

Comments on the Quality of English Language

Some areas need editing on English.

Round 2

Reviewer 1 Report

Comments and Suggestions for Authors

Following the revision, the manuscript's quality has significantly improved.

Author Response

Thank you for your review.

Reviewer 2 Report

Comments and Suggestions for Authors

Peer review:

Manuscript “Swine acute diarrhea syndrome coronavirus: An updated overview of virus‒host interactions” (animals-3364235R1)

Review comments to the authors:

As I previously mentioned in my first analysis, the manuscript is a review on the swine acute diarrhea syndrome coronavirus (SADS-CoV). The review focuses on the characteristics of the virus and its interactions with the host. It also addresses potential therapeutic molecules and vaccines to prevent SADS-CoV infection.

I have previously pointed out that the manuscript was incomplete and described some necessary adjustments. The authors have improved the manuscript by including a specific section on the epidemiology of SADS-CoV, detailing the spread and molecular characterization of the different strains involved in previous outbreaks (i.e., writing in more detail and in a separate section what is summarized in the second paragraph). However, this section is written in just one confusing paragraph. I would recommend further effort to better describe the epidemiology of the major swine pathogenic coronaviruses associated with major outbreaks worldwide. Figure 1 also needs to be improved (or at least better explained), since it is presenting coronaviruses from different hosts. I strongly recommend the construction of a more informative phylogenetic tree, highlighting the major swine coronaviruses from the different viral genera .  

The authors have also made most of the other minor adjustments that were needed. I am only additionally recommending that the abbreviations used be explained. For example, what does “HKU-2” (in the first sentence of the Simple Summary) mean? Remember that the meaning of each abbreviation needs to be explained the first time it is used.

Author Response

Comment 1: I have previously pointed out that the manuscript was incomplete and described some necessary adjustments. The authors have improved the manuscript by including a specific section on the epidemiology of SADS-CoV, detailing the spread and molecular characterization of the different strains involved in previous outbreaks (i.e., writing in more detail and in a separate section what is summarized in the second paragraph). However, this section is written in just one confusing paragraph. I would recommend further effort to better describe the epidemiology of the major swine pathogenic coronaviruses associated with major outbreaks worldwide. Figure 1 also needs to be improved (or at least better explained), since it is presenting coronaviruses from different hosts. I strongly recommend the construction of a more informative phylogenetic tree, highlighting the major swine coronaviruses from the different viral genera.  

Answer 1: Thank you for your comments. To date, six swine CoVs have been identified. These include transmissible gastroenteritis virus (TGEV), porcine epidemic diarrhea virus (PEDV), porcine respiratory coronavirus (PRCV), porcine hemagglutinating encephalomyelitis virus (PHEV), porcine acute diarrhea syndrome coronavirus (SADS-CoV), and porcine delta coronavirus (PDCoV). It may be valuable to describe the epidemiology of the major swine pathogenic coronaviruses (especially PEDV) associated with major outbreaks worldwide. However, our review focuses on recent developments in the SADS-CoV outbreak, so we will not include this information. Furthermore, this information has already been reviewed in other review articles. Five CoVs, except PRCV, were included in Figure 1. Based on your comments, all six swine CoVs were included in the revised Figure 1 and highlighted in blue.

Comment 2: The authors have also made most of the other minor adjustments that were needed. I am only additionally recommending that the abbreviations used be explained. For example, what does “HKU-2” (in the first sentence of the Simple Summary) mean? Remember that the meaning of each abbreviation needs to be explained the first time it is used.

Answer 2: We have added additional explanations for acronyms. HKU-2 is not an acronym, it is a strain of Rhinolophus bat coronavirus. The manuscript has been revised to clarify this.